



# Examining baseline water properties and bottom water patterns in hadal trench environments

Jessica Kolbusz[1], Jan Zika[2], Charitha Pattiarachi[3] and Alan Jamieson[1]

[1]Minderoo-UWA Deep-Sea Research Centre, University of Western Australia, Nedlands, 6009, Australia
[2]School of Mathematics and Statistics, University of New South Wales, Sydney, 2052, Australia
[3]Oceans Graduate School, University of Western Australia, Nedlands, 6009, Australia

*Correspondence to*: Jessica L. Kolbusz (jess.kolbusz@uwa.edu.au)

**Keywords**

Antarctic Bottom Water, hadal, hadopelagic, Lower Circumpolar Deep Water, trench

**Abstract**

We examine baseline water properties and bottom water patterns in hadal trench environments across the Southern Ocean, Indian Ocean, and Western Pacific. Significant differences are identified in the South Fiji Basin and surrounding the Philippine Sea, primarily due to the movement of cold Lower Circumpolar Deep Water along topographic features, highlighting the importance of a trench's geospatial position. We present the first hydrographic profiles in the Java Trench, warranting further
research. Increases in salinity patterns in depths over 10,000 dbar are investigated, with potential causes including instrumentation error, internal mixing, and saline pore water expulsion. These hadopelagic variations are crucial for assessing climate change impacts, especially regarding Antarctic Bottom Water. The study underscores the importance of incorporating these adiabatic conditions for insights into ecological biodiversity, alongside the baseline conditions presented being indispensable for future oceanographic research across multiple disciplines.

**1 Introduction**

The oceanic hadal zone, defined as regions where water depths are greater than 6,000 m, is located mainly in subduction trenches, fracture zones and basins (Jamieson & Stewart, 2021). This remote region of water is arguably the most understudied habitat in the marine environment (Webb et al., 2010). The most conspicuous geomorphological features of the hadal zone are the large subduction trenches in the Atlantic, Southern, Indian and, particularly, the Pacific Ocean (Jamieson, 2015). These
trenches are characterised by extreme pressures, cool temperatures, limited light penetration and frequent subduction earthquakes (Jamieson et al., 2010; Taira et al., 2005; Oguri et al., 2013). Of these 27 trenches, most exceed 8,000 m in depth, with four exceeding 10,000 m (Bongiovanni et al., 2021). While the total 2-dimensional area of the seafloor greater than 6,000 m is likely less than 1% (Harris et al., 2014), the volume of pelagic waters contained within these enclosed topographies accounts for 45% of the total ocean depth range (Jamieson, 2015).





Sampling and exploration of the hadal benthic habitats have, in recent years, seen a renaissance (Jamieson, 2018), with biological data sets being obtained for multiple taxonomic groups from multiple depths over multiple trenches (Jamieson, Linley, et al., 2021; Jamieson & Linley, 2021; Swan et al., 2021; Weston & Jamieson, 2022). As these studies develop from simple species discovery and observations of presence/absence, other abiotic factors are required to explain larger-scale ecological patterns in vertical distribution and biogeography (Nunoura et al., 2016; Glud et al., 2013). For example, the hadal

benthic fauna are supported by the deposition of nutritious organic material from the surface ocean into the trench sediments (Ichino et al., 2015), increasing microbial activity and carbon accumulation through mass-wasting events or internal waves (Oguri et al., 2022; Turnewitsch et al., 2014; van Haren, 2020b). In addition, understanding these habitats' geomorphological structuring is valuable in explaining species' relationships with substrate (Stewart and Jamieson, 2018; Jamieson et al., 2022). However, hydrographic conditions such as temperature and salinity and inter-trench variation therein are seldom accounted

for in observations (Jamieson and Fujii, 2011).

While single point temperature and salinity measurements were made in multiple trenches in the 1950s and 60s (Belyaev, 1989), full-ocean-depth hydrographic profiles of conductivity (for salinity), temperature and depth (CTD) are rarely reported, excluding those surrounding the Izu-Bonin-Mariana Arc System (Stewart and Jamieson, 2018). Adiabatic heating through the water column (Bryden, 1973) is often reported in ecological contexts to explain why ambient bottom temperatures at hadal

depths are warmer than at abyssal depths (Jamieson et al., 2010; 2021; Jamieson & Stewart, 2021). With the prestige of being the deepest point on Earth, most CTD profiles are biased towards the *Challenger Deep* in the Mariana Trench (Greenaway et al., 2021; Mantyla and Reid, 1978; Taira et al., 2005; van Haren et al., 2017, 2021), and neighbouring trenches (Kawagucci et al., 2018; Taira, 2006). The same applies for long-term deployments of hydrographic moorings (Taira et al., 2004; van Haren, 2020a). The combination of the Mariana Trench prestige and oceanographic sensor depth limitations of 6000 m, have led to

few studies addressing water column characteristics and comparisons between trenches over the hadopelagic.

In the broader context of the global overturning circulation, trenches are filled by, or may serve as conduits for the transport of dense bottom water masses. Temperature and salinity gradients drive the overturning circulation, which transports cool, dense water, formed around Antarctica (Antarctic Bottom Water, AABW) in the Weddell Sea, Ross Sea and Adélie Land (Zhang and Delworth, 2016; Talley, 2013a; Gordon, 1986b; Rintoul, 1985) as well as North Atlantic Deep Water (NADW),

formed from dense overflows in the Nordic Seas, lower Labrador Sea Water and recirculating AABW, to fill the ocean basins (Gordon, 1986a; Dickson and Brown, 1994; Orsi et al., 1999). The cool water eventually rises, and is heated, flowing poleward to replace denser water, creating a global cell of circulation (Stommel and Arons, 1959). Diapycnal exchanges also play an important role in modifying pathways and ventilation timescales (Cimoli et al., 2023). However, the full implication of these findings and how they play into the variation in trench conditions is yet to be understood on an inter-basin scale.

The Southern Ocean plays a significant role in the deep overturning circulation both in formation of AABW and its influence on NADW production and cell closure (Rahmstorf and England, 1997; Goodman, 1998; Sloyan and Rintoul, 2001). This lower limb of the overturning circulation fills the abyssal basins of the Atlantic, Indian and Pacific Oceans (Orsi et al., 1999, 1993; Mantyla and Reid, 1995). AABW upwells locally as it moves northwards, making it the primary volumetric source of deep





water in the Pacific and Indian Oceans, where there is no high-salinity water source to the north (Talley, 2011). These patterns
of sinking and spreading through the world's oceans have been captured in observations (Wust, 1933) and are considered a
key component of climate regulation (Naveira Garabato et al., 2017; Schmitz, 1995).

In the Pacific Ocean, Lower Circumpolar Deep Water (LCDW) mostly comprises of AABW and remnant NADW from the
Southern Ocean (Johnson, 2008; Orsi et al., 1999). LCDW flows northward on the western side of the Southwest Pacific Basin,
the Tonga-Kermadec Ridge (TKR), and reaches the Samoan Passage near the northern end of the Tonga Trench (Roemmich
et al., 1996; Taft et al., 1991). The TKR prevents abyssal water from entering the South Fiji Basin; therefore bottom waters
are formed by the densest parts of the Upper Circumpolar Deep Water (UCDW) entering the New Hebrides Basin via the
entrance to the southern Solomon Sea Basin (Germineaud et al., 2021; Kawabe and Fujio, 2010).

Just north of the Samoan Passage, the LCDW then bifurcates into a western and eastern branch driven by the complex
topography in the region (Kawabe et al., 2003). The western branch flows north-westward, arriving in the East Mariana Basin
where it separates into East Caroline Basin to the south and north, and is transported to the Izu-Ogasawara Trench followed
by the Japan Trench (Fujio et al., 2000; Fujio, 2005; Siedler et al., 2004). A portion of the LCDW reaches the south of the
Mariana Trench, the Yap-Mariana Junction from the northward flow (Kawabe and Fujio, 2010), where it flows into the
northern and southern Yap Trench and partially into the Philippine Basin (Liu, 2022; Liu et al., 2020; Zhai and Gu, 2020). The
Yap Ridge then blocks the circulation of water causing it to enter to the Western Caroline Basin, where it is able to enter the
Palau Trench (Zhai and Gu, 2020; Liu, 2022). Modelling suggests that the bottom layers of the Philippine Basin form an
anticyclonic gyre in the south and two cyclonic gyres in the northeast and northwest, followed by possible cyclonic circulation
over the Philippine Trench (Zhai and Gu, 2020).

Bottom waters in the Indian Ocean are dominated by AABW as it enters through two pathways and is observed as far north as
12°N in the Bay of Bengal (Chinni et al., 2019; Singh et al., 2012). Firstly, it enters through the Enderby Basin, having formed
in the Weddell Sea (Orsi et al., 1999). Secondly, through the Australian-Antarctic Basin, where the bottom water was originally
formed from the Adélie Land and the Ross Sea (Rintoul, 1985; Mantyla and Reid, 1995). Elevated dissipation rates from tidal
mixing and current fluctuations in the Indian Ocean contribute to a higher ventilation rate of deep and bottom water in the
Indian Ocean in comparison to the Pacific and Atlantic Oceans (Fine et al., 2008). The deep waters of the Indian ocean,
similarly to in the Pacific Ocean, are sourced primarily from upwelled bottom waters, which are then heated and, through
diapycnal transformation, returned to the sea surface, and are cooled and recycled back into AABW in the Southern Ocean
(Talley, 2013a).

In this paper, we explore twelve hadal trenches using full-ocean-depth CTD profiles derived from free-fall landers to unravel
the complexities of the trenches and the interplay between the hadopelagic and global circulation patterns of bottom waters.
The World Ocean Circulation Experiment (WOCE) measured vertical sections along each hydrographic line covering the
globe in the late 1980s and early 2000s, with some repeat sections (Koltermann et al., 2011; Orsi & Whitworth, 2005; Talley,
2007, 2013a). Measurements were restricted to 6000 m, although global transport estimates were substantial for climate studies
(Ganachaud, 2003; Ganachaud and Wunsch, 2000). By analysing the opportunistic full-ocean-depth CTD data sets, alongside



repeat WOCE global observations, we aim to gain insights into hadal trench connections with the abyssal waters above.
Through this analysis, we describe physical oceanographic conditions within the hadal trenches of the Southern Ocean, Indian

Ocean and Pacific Ocean, spanning seven of the ten hadal provinces (Watling et al., 2013). Furthermore, we link changes in
water properties in the western Pacific Ocean to AABW and LCDW source waters to enhance our understanding of the physical
oceanographic conditions which shape ecosystem survival in these unique environments.

**Figure 1. Geographic locations of the hadal trenches used in this study, including CTD deployment locations marked in black. (a)
Ryukyu Trench and Philippine Trench; (b) Japan Trench and Izu-Ogasawara Trench; (c) Palau Trench, Yap Trench, and Mariana
Trench; (d) Java Trench (e) Santa Cruz Trench and New Hebrides Trench; (f) Tonga Trench and Kermadec Trench and (g) South**





**Sandwich Trench. Individual deployment details are provided in Table S1. Bathymetric details and features are provided in Table S1. Ocean elevation data was sourced from the GEBCO Compilation Group (2021) and reproduced using m_map in MATLAB (Pawlowicz 2020).**


## 2 Methods

We define the hadal water properties and bottom conditions of twelve of the world's hadal trenches spanning three oceans and seven hadal biogeographic provinces (Belyaev, 1989; Watling et al., 2013) (Table 1). Full-ocean-depth landers mounted with CTDs were deployed between 2018 and 2022. The deployment locations, relevant features and nearby WOCE sections

mentioned throughout the paper are indicated in Figure 1. The location of features mentioned, other than the trenches are found in Table S1. Quality control and analysis are detailed in the following section, producing 33 CTD profiles in total (Figure 1, Table 1, Table S1).

### 2.1 Study sites

In the Southern Ocean, deployments were made in the South Sandwich Trench (SAND) (~55˚S / 23˚W) to 8,266 m. SAND

falls within the Southern Antilles hadal province (Watling et al., 2013). Bottom circulation in the region is promoted by the Weddell Gyre exporting of newly formed dense Weddell Sea Bottom Water, or Antarctic Bottom Water (AABW), northward through the Scotia Sea and along the South Sandwich Trench through the steep slopes inducing strong downwelling (Zhang and Delworth, 2016).

The deepest point of the Indian Ocean is the Java Trench (JAV), where deployments were made to 7,178 m (11˚S / 115˚E).

Bottom water (modified AABW and LCDW) reaches the Java Trench from the West Australian Basin to the south (Mantyla and Reid, 1995; Sloyan, 2006; Arvapalli. et al., 2022).

In the central South Pacific, deployments were made in the Tonga Trench (~23˚S / 174˚W) to a maximum depth of 10,823 m, 9,986 m and in the Kermadec Trench (~32˚S / 177˚E); The Kermadec Trench connects at the southern end of the Tonga Trench within the same convergence system, and they are separated only by the subducting Osborn Seamount (Jamieson et al., 2020).

Bottom circulation in the region is dominated by the major AABW/LCDW pathway into the Pacific, which enters from the south along the Tonga-Kermadec Ridge and flows north (Warren, 1981; Kawabe, 1993; Taft et al., 1991; Purkey et al., 2019). Surrounding the Solomon Sea lies the Bougainville-New Hebrides hadal province, where deployments were made to 7,770 m in the New Hebrides Trench (23˚S / 172 ˚E) and 9,125 m in the Santa Cruz Trench (11˚S / 163 ˚E). Bottom water masses in these trenches are akin to Upper Circumpolar Deep Water (UCDW) which have entered the Solomon Sea from the north

(Germineaud et al., 2021).

In the central northwest Pacific Ocean, deployments were completed to a maximum of 10,925 m in the Mariana Trench (MAR) (~11˚N / 142˚E); 8,885 m in the Yap Trench (YAP) (~9˚N / 138˚E); 8,000 m in the Palau Trench (PAL) (~8˚N / 135˚E) and, 10,007 m in the Philippine Trench (PHI) (~10˚N / 126˚E). Further north, there were deployments to 9,767 m in the Izu Ogasawara Trench (IOT) (29˚N / 142˚E) and to 8,004 m in the Japan Trench (JPT) (36˚N / 143˚E). These deployments span



three hadal provinces (Table 1). Bottom water circulation in this region is driven by AABW/LCDW, which enters the east of
the Mariana Basin in the North Pacific (Kawabe et al., 2003; Siedler et al., 2004). The westward component propagates through
MAR, flowing through the westernmost end and over the YAP and PAL trenches (Kawabe and Fujio, 2010). The northern
branch flows through the Wake Island Passage and along the Izu-Ogasawara Ridge, filling IOT and JPT (Tian et al., 2021).
The Philippine hadal province has a unique location west of the Philippine Basin, not immediately near the primary bottom
water circulation of the Pacific Ocean (Kawabe and Fujio, 2010). Deep bottom circulation along the trench consists of
southward and northward currents on the western and eastern slopes of the trench, respectively (Zhai and Gu, 2020), which is
not unlike other local cyclonic circulation observed in other trenches (Huang et al., 2018). This deep water mass is likely well-
transformed AABW/LCDW that fills the Philippine Sea from the north (Wang et al., 2017; Kawabe and Fujio, 2010; Tian et
al., 2021).


**Table 1. Summary of the hadal lander deployments from RV Pressure Drop used in this study. Specific deployment details in Table S1.**

| Trench | Hadal Province (Belyaev, 1989; Watling et al., 2013) | # Profiles | Date | Depth range (m) | WOCE Section |
|---|---|---|---|---|---|
| South Sandwich Trench (SAND) | Southern Antilles | 2 | February 2019 | 8,071-8,254 | S4 |
| Java Trench, or Sunda Trench (JAV) | Java | 3 | April 2019 | 6,136-7,197 | I10, I06 |
| Kermadec Trench (KRT) | Tonga-Kermadec | 1 | December 2021 | 9,978 | P06 |
| Tonga Trench (TON) | Tonga-Kermadec | 2 | June 2019 | 7,471-10,811 | P15 |
| New-Hebrides Trench (NHT) | Bougainville-New Hebrides | 1 | December 2021 | 7,960 | P21 |
| Santa Cruz Trench (SCZ) | Bougainville-New Hebrides | 1 | December 2021 | 9,370 | P21 |
| Mariana Trench (MAR) | Mariana | 5 | April 2019 and June 2020 | 7,495 – 10,922 | P04 |
| Yap Trench (YAP) | Mariana | 2 | July 2022 | 5,981 – 8,885 | P04, P09 |
| Palau Trench (PAL) | Mariana | 1 | July 2022 | 7,995 | P04, P09 |
| Philippine Trench (PHT) | Philippine | 3 | February 2021 | 6,995- 10,058 | P04, P08 |
| Izu-Ogasawara Trench (IOT) | Aleutian- Japan | 6 | August 2022 | 6,462 – 9,752 | P02, P09 |
| Japan Trench (JPT) | Aleutian- Japan | 5 | September 2022 | 6,462 – 7,991 | P10, P09 |



## 2.2 Data collection

The hydrographic CTD profiles were collected as part of the *Five Deeps Expedition* and the follow-on *Ring of Fire Expedition* (2020-2022), on the RV *Pressure Drop*. To support the diving operations of the full-ocean-depth submersible DSSV *Limiting Factor*, and collect scientific data (e.g. Jamieson et al., 2021), three hadal landers were used in unison (known as *Skaff, Flere* and *Closp*). The landers were equipped with CTD probes (SBE 49 FastCAT; SeaBird Electronics, Bellevue, US) that, during both the descent and ascent, recorded conductivity, temperature, and pressure at 10-second intervals. The CTDs measured

conductivity, temperature and pressure at ± 0.0003 S/m, ±0.002ºC and ±0.1% of the full-scale range, respectively (Sea-Bird Scientific, 2020).

The landers were released from the ship and descended until reaching the seafloor, where they performed autonomous filming operations for several hours before their ascent was initiated by jettisoning ballast weights via acoustic command. The landers descended at an average speed of 0.6 ms$^{-1}$, and the design did not require any corrections for ship motions in CTD casts.

## 2.3 Data processing

The CTD data were processed using the standard procedures incorporated within the SBE Data Processing Software (Sea-Bird Scientific, 2018). Automated processing including aligning the temperature and conductivity data in time relative to the pressure and cell thermal mass corrections occurred within the sensor (Sea-Bird Scientific, 2020). A Wild Edit flagged data outside two standard deviations away from the mean. This was followed by applying a median filter to remove spikes. These

methods were completed as recommended by Seabird. Profiles were discarded if Wild Edit bad flags were greater than 20% of the profile or if other lander-related situations occurred, influencing the data, and making the profile unusable. Post-cruise calibrations were carried out, as recommended by SeaBird and post-corrections were made using offset correction coefficients. Absolute salinity ($S_A$, g kg$^{-1}$), Conservative Temperature ($\Theta$) and Potential Density referenced to 6000 dbar ($\sigma_6$) were calculated using the Gibbs-SeaWater (GSW) software which defines the thermodynamic properties of seawater based on a Gibbs function

formulation as recommended by IOC (IOC et al., 2010). Then, the down-cast profiles were averaged to 10 dbar bins with the first 50 dbar of the water column removed due to lander effects and a gaussian filter applied.

A conservative approach was taken with the data due to limited profiles and possible limitations, including the drift of conductivity measurements in SBE CTDs, (van Haren et al. 2017; Goretski et al. 2022). Therefore, if there was a World Ocean Circulation Experiment (WOCE) station deeper than 4500 m, taken within ±4 years of the lander deployment and lying within

±8° longitude and ±3° latitude of its location, a salinity offset from the WOCE station was applied to the CTD data (Supplementary 3). In total, 33 CTD profiles were used (Figure 1).

## 2.3 Data analysis

We derive the comparative mean water conditions together with $\Theta$-$S_A$ plots and apply a source water type analysis. Through these mechanisms we are addressing the heterogeneity of trenches. The downcast profiles were used to eliminate the potential



heating effects generated by illumination devices on the landers. Only profiles measuring to over 6000 m were considered in this study. Despite our focus being on the hadopelagic (> 6000 m), measurements within abyssal waters between 4000 and 6000 m are discussed and used in conjunction with WOCE data for baseline and temporal and spatial comparisons (Figure 1). $\Theta$-$S_A$ plots, derived initially by Helland-Hansan (1916), were then plotted with isopycnals referenced to 6000 dbar (Pond and Pickard, 1983). This reference pressure was adopted given the depths of the profiles (> 6000 dbar), therefore more accurately

reflecting the thermodynamic properties of deep-water masses.

The mean in-situ Temperature, T, $\Theta$ and $S_A$ at 7000 m were calculated in each trench for comparison since the Java Trench is the shallowest included within this study at 7192 m (Bongiovanni et al., 2021). A $\Theta$-$S_A$ plot of bottom conditions in the trenches was also generated for comparative purposes.

We apply a source water type analysis to the profiles along the western Pacific, likely containing Antarctic Bottom Water

(AABW) and Lower Circumpolar Deep Water (LCDW) to understand the broad spatial differences. Restrictions on using basic Optimum multiparameter analysis (OMP) at a basin scale meant that mixing, including biogeochemical processes, cannot be ignored (Karstensen and Tomczak, 1998; Leffanue and Tomczak, 2004), and in our instance, we were limited to the conservative parameters of temperature and salinity. Trenches included were KRT, TON, MAR, YAP, PAL, PHI, IOT and JPT. The source water types of AABW and LCDW were defined along neutral density ($\gamma_n$) isopycnals, $\gamma_n > 28.2$ kg m$^{-3}$ and

28.0 $\gamma_n < 28.2$ kg m$^{-3}$, respectively (Table 2), at 60°S along the World Ocean Circulation Experiment P15 line (Figure 1) (Tamsitt et al., 2018; Abernathey et al., 2016; Solodoch et al., 2022).

We categorize the observations of conservative temperature ($\Theta_{Obs}$) and absolute salinity ($S_{Obs}$) as being a mixture of two source water masses, AABW and LCDW. To estimate the fractional contributions of these two source waters ($x_{AABW}$ and $x_{LCDW}$ both

between 0 and 1) we seek a solution to the following equations, inspired by classical OMP analysis (Tomczak, 1999; Tomczak and Large, 1989).

$$x_{AABW}\Theta_{AABW} + x_{LCDW}\Theta_{LCDW} = \Theta_{Obs}$$
$$x_{AABW}S_{A_{AABW}} + x_{LCDW}S_{A_{LCDW}} = S_{A_{Obs}}$$
$$x_{AABW} + x_{LCDW} = 1$$

We solve for $x_{AABW}$ and $x_{LCDW}$ for each observation by performing matrix left division in MATLAB. As the fractions were constrained to total 1, only the AABW fraction is reported in the results. To account for uncertainty in the source water properties the procedure was repeated for four combinations of maximum and minimum bounds of AABW and LCDW $\Theta$ and

$S_A$ (Table 3).

**Table 2. Source water type values calculated from WOCE section P15.**



|  | Conservative Temperature ($\Theta$) °C | Absolute Salinity ($S_A$) g kg$^{-1}$ |
|---|---|---|
| **AABW$_{min}$** | 0.1115 | 34.8645 |
| **AABW$_{max}$** | 0.1119 | 34.8639 |
| **LCDW$_{min}$** | 1.2812 | 34.8863 |
| **LCDW$_{max}$** | 1.4526 | 34.8993 |

**Table 3. Source water type combinations used to get the range uncertainty in the calculated source water type values defined in Table 2.**

|  | Source water type combination |
|---|---|
| **SWT1** | AABW$_{min}$ and LCDW$_{min}$ |
| **SWT2** | AABW$_{min}$ and LCDW$_{max}$ |
| **SWT3** | AABW$_{max}$ and LCDW$_{min}$ |
| **SWT4** | AABW$_{max}$ and LCDW$_{max}$ |

# 3 Results

The full-ocean-depth CTD profiles were limited to over 4000m and included in-situ temperature (T), practical salinity ($S_P$) and pressure (P). From these, conservative temperature ($\Theta$) and Absolute Salinity ($S_A$) from the 12 hadal trenches were analysed to describe the conditions in the trenches and bottom conditions. Interpretation of $\Theta$-$S_A$ plots for each study region and additional profile analysis for trenches reaching 10,000 decibar was then completed.

## 3.1 Water Properties

In the southernmost trench, the South Sandwich Trench (SAND), the hadal water mass characteristics were similar between the two deployments despite being 2º of latitude apart (Figure 1). From 4000 dbar to the hadopelagic, $\Theta$ decreased by 0.15ºC, while $S_A$ had negligible vertical variation (Figure 2). The WOCE section, S4, decreases monotonously in $\Theta$ and $S_A$ over depth, reaching a cooler bottom temperature of -0.76ºC.

In the Java Trench, otherwise known as the Sunda Trench, the average d$\Theta$/dp was -0.05ºC/1000 dbar, and d$S_A$/dp was 0.0013 g kg$^{-1}$/1000 dbar between 4000 and the seafloor (7360 dbar) (Figure 3). There is a more rapid increase toward the seafloor, which is also evident in section I06.

The profile from KRT is consistent with P06 stations nearby; however, there is an increase in $S_A$ of 0.002 g kg$^{-1}$ in the bottom 1000 dbar of the trench (Figure 4b). Two of the profiles from TON indicate a similar increase in $S_A$ at the bottom 1000 dbar of 0.015 g kg$^{-1}$. The remaining profile increases slightly in $S_A$ (0.001 g kg$^{-1}$), following P15 more closely; however, it reaches only 6800 dbar (Figure 4a). The abyssal depths of WOCE sections P06 and P15 passing near TON and KRT are discussed extensively in the literature (Katsumata and Fukasawa, 2011; Macdonald et al., 2009). The profiles near TON and KRT decrease monotonously in $\Theta$ and $S_A$ from 4000 dbar to the seafloor.



In the New Hebrides Trench (NHT), $d\Theta/dp$ was -0.006ºC/1000 dbar and $dSA/dp$ was 0.0056 g kg$^{-1}$/1000 dbar between 4000 and the seafloor (7940 dbar). The rate of change was marginally larger in the Santa Cruz Trench (SCZ). $D\Theta/dp$ was -0.008ºC/1000 dbar and $dS_A/dp$ was 0.008 g kg$^{-1}$/1000 dbar between 4000 and the seafloor (9350 dbar) (Figure 5).

In the Palau Trench (PAL), $d\Theta/dp$ was -0.018ºC/1000 dbar and $dSA/dp$ was 0.0013 g kg$^{-1}$/1000 dbar between 4000 and the seafloor (8190 dbar) (Figure 6a). In the Yap Trench (YAP) the average rate of change in $\Theta$ was -0.055ºC/1000 dbar and $dS_A/dp$

was 0.004 g kg$^{-1}$/1000 dbar between 4000 and the seafloor (8190 dbar). The two YAP profiles diverge in $S_A$ at 5200 dbar with the deeper and more northern profile increasing in $S_A$ at a greater rate into the trench (Figure 6). In the Mariana Trench (MAR), $d\Theta/dp$ is $-2.16 \times 10^{-5}$ ºC/1000 dbar due to near isothermal conditions over the large depth between 4000 dbar and the seafloor (~11240 dbar). The rate of change of $S_A$ is also low ($2.64 \times 10^{-6}$ g kg$^{-1}$/1000 dbar) due to isohaline conditions; however, we observed an increase of ~0.01g kg$^{-1}$ in the bottom 1500 dbar of the four profiles exceeding 10800 dbar (Figure 6b).

The profiles in the Izu-Ogasawara Trench, $d\Theta/dp$ was -0.024ºC/1000 dbar and $dS_A/dp$ was 0.0026 g kg$^{-1}$/1000 dbar between 4000 and the seafloor (maximum of 10010 dbar) (Figure 7a). An increase in $S_A$ of ~0.002 g kg$^{-1}$ in the bottom 1000 dbar of the deepest profile was observed. In the Japan Trench (JPT), $d\Theta/dp$ was -0.033ºC/1000 dbar, and $dS_A/dp$ was 0.0026 g kg$^{-1}$/1000 dbar between 4000 and the seafloor (maximum of 8170 dbar) (Figure 7a).

In the Philippine Trench (PHI), $d\Theta/dp$ was -0.005ºC/1000 dbar, and $dS_A/dp$ was 0.0005 g kg$^{-1}$/1000 dbar between 4000 and

the seafloor (maximum of 10340 dbar) (Figure 8a). An increase in $S_A$ of ~0.003 g kg$^{-1}$ was observed in the bottom 1500 dbar of the deepest profile (Figure 8b).

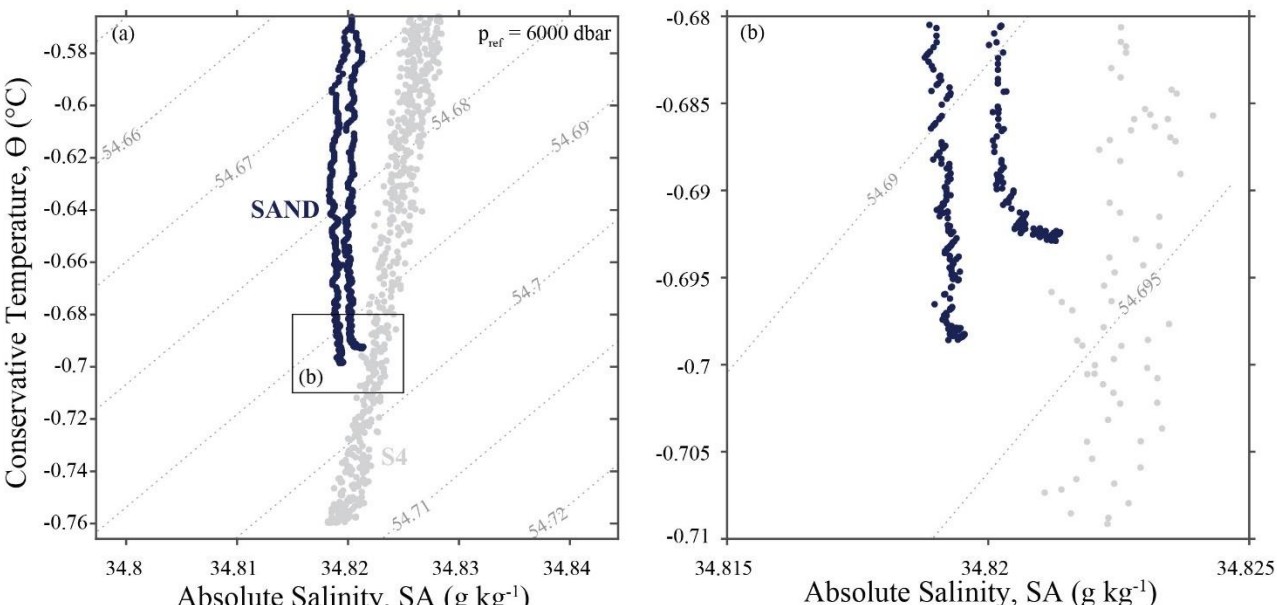

**Figure 2. Ө-SA plots for the South Sandwich Trench (SAND) (a) over 4000 dbar (b) bottom waters. The rectangle in (a) shows the extent of (b). Isopycnals (contour lines) show the potential density referenced to 6000 dbar. The grey data points are WOCE section**
**S4.**



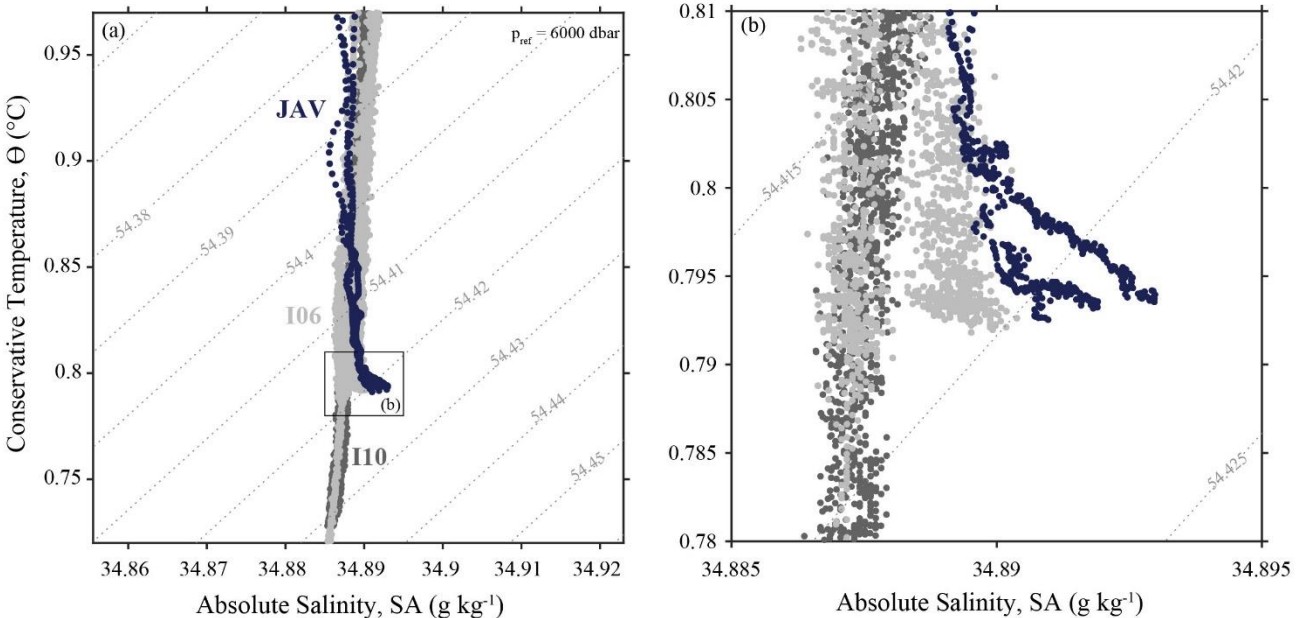

**Figure 3. As in Figure 2 for the Java Trench. The grey is WOCE section I06 and I10.**

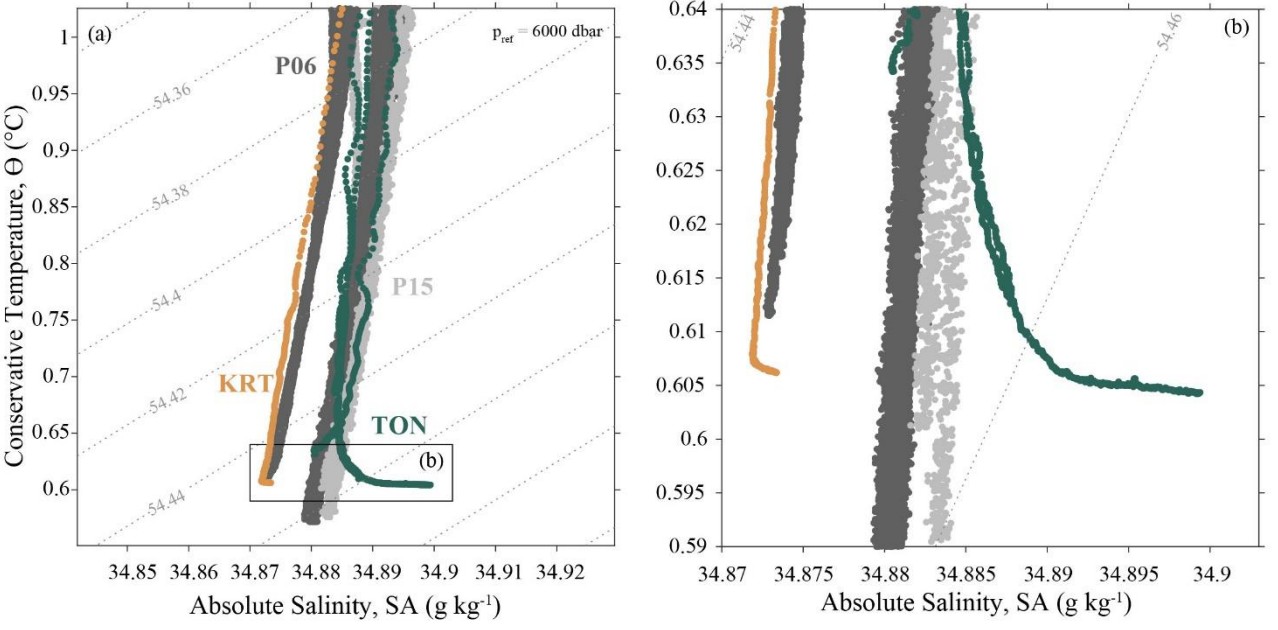

**Figure 4. As in Figure 2 for the Kermadec Trench (KRT) and Tonga Trench (TON). The grey is WOCE section P06 and P15.**



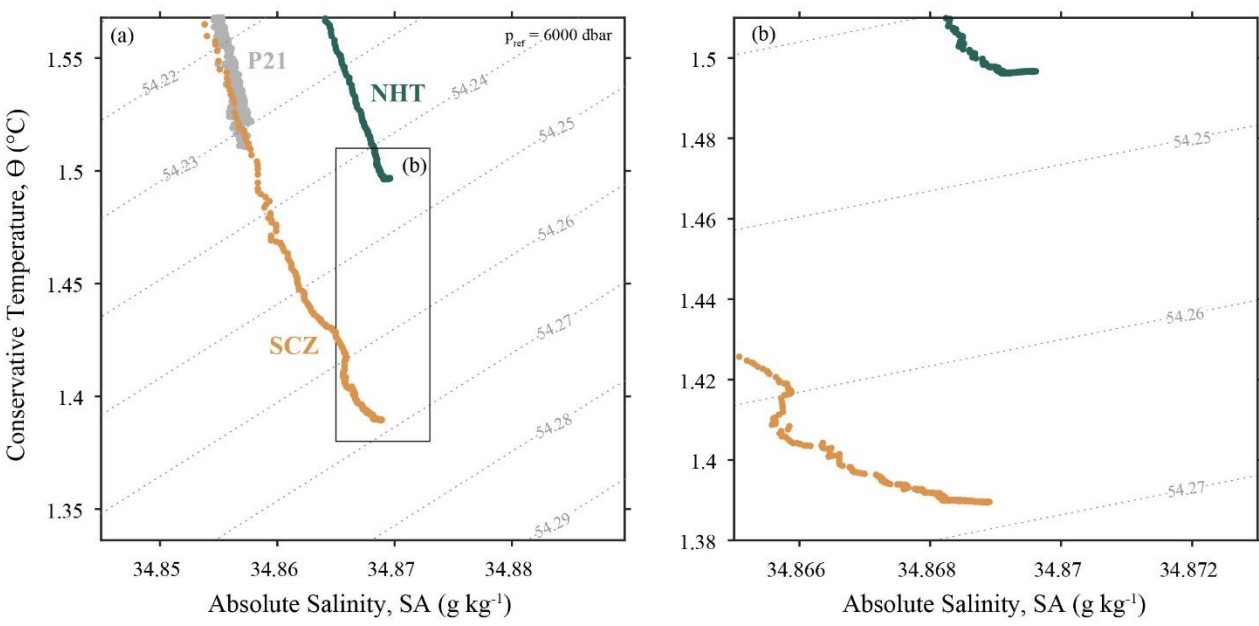


**Figure 5. As in Figure 2 for the Santa Cruz Trench (SCZ) and New Hebrides Trench (NHT). The grey is WOCE section P21.**

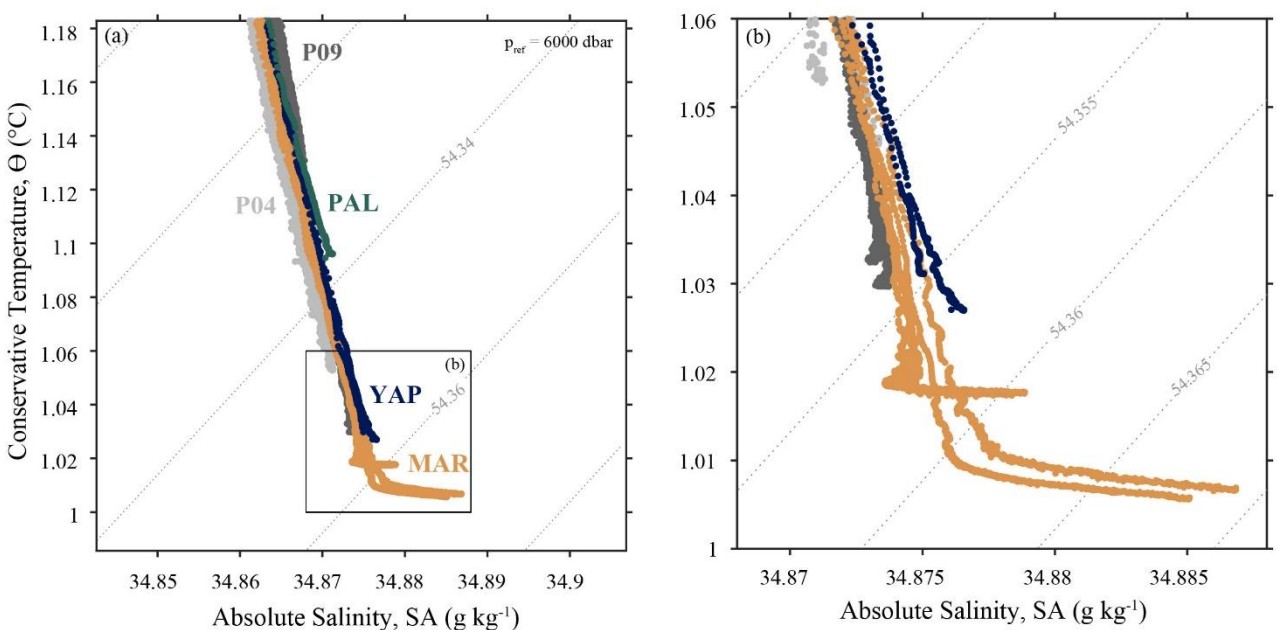

**Figure 6. As in Figure 2 for the Mariana Trench (MAR), the Palau Trench (PAL) and Yap Trench (YAP). The grey is WOCE section P04 and P09.**



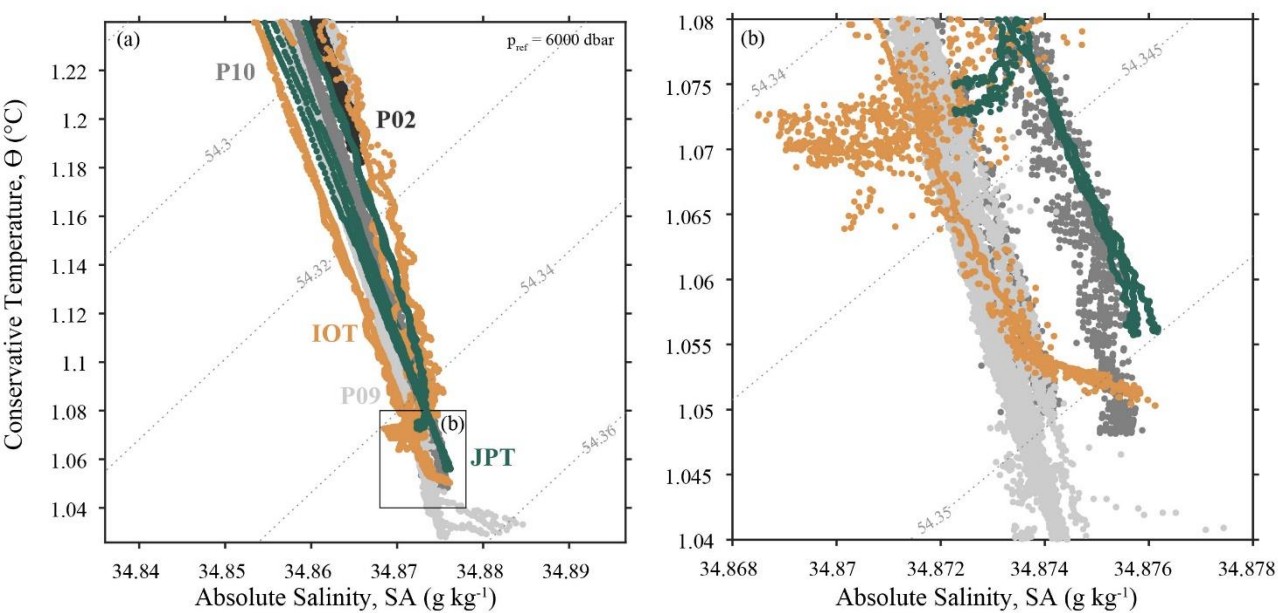

**Figure 7. As in Figure 2 for the Japan Trench (JPT) and Izu-Ogasawara Trench (IOT). The grey is WOCE section P09 and P10**

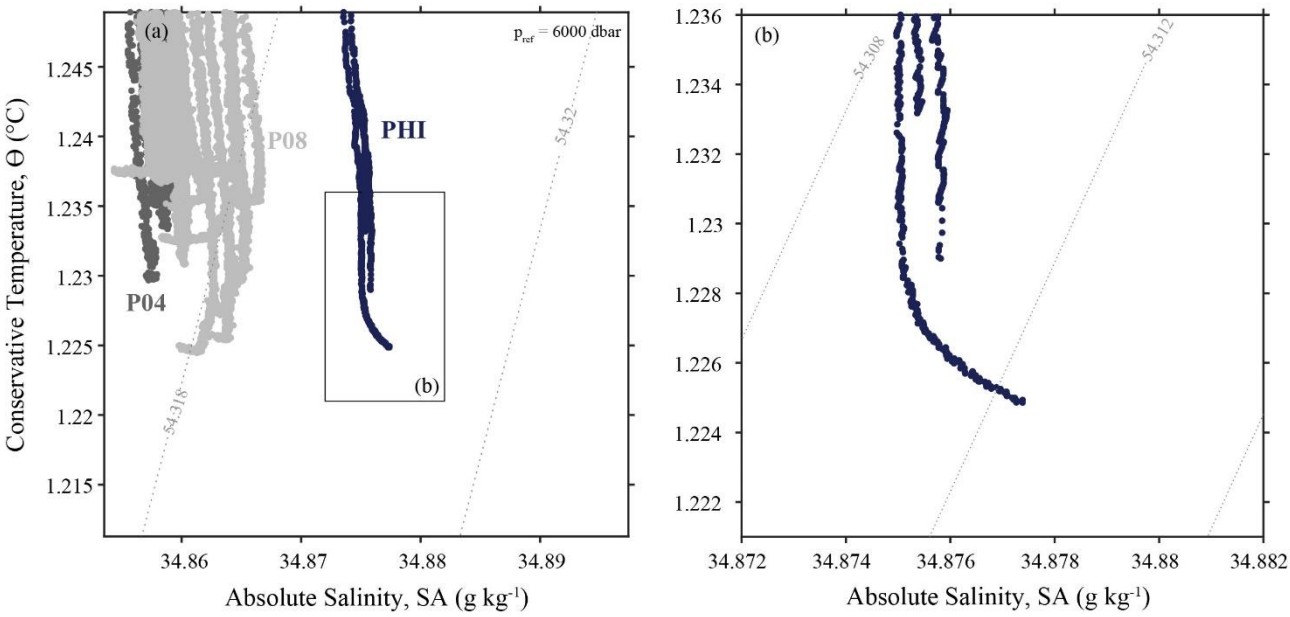

**Figure 8. As in Figure 2 for the Philippine Trench. The grey is WOCE section P04 and P08**



## 3.2 Bottom Conditions

The South Sandwich Trench (SAND) is unique in its location in the Southern Ocean and proximity to bottom water formation compared to the other trenches. It is the coolest ($\Theta$ = 0.69ºC, In-situ Temperature, T = 0.03ºC) and most fresh (SA = 34.82 g kg$^{-1}$) of the trenches observed (Figures 9 and 10). The Java Trench (JAV) is the only observation within the Indian Ocean and

is the most saline of the twelve trenches at 7000 dbar ($S_A$ = 34.89 g kg$^{-1}$) (Figure 9c). The trenches become warmer from south to north along the western Pacific (Figures 9a and c). This excludes NHT and SCZ within the Bougainville-New Hebrides hadal province, which exhibited warmer and fresher water properties than TON-KRT to the east and MAR-YAP-PAL to the north. Within the northwest Pacific, the Philippine Trench (PHI) is the warmest of the trenches surrounding the Philippine Basin in terms of bottom temperature ($\Theta$ = 1.24ºC) (Figure 10). The average bottom conditions of the trenches within the same

hadal province are within the same space in $\Theta$-$S_A$ space, with the Mariana and Aleutian-Japan hadal provinces overlapping (Figure 10). These results suggest an association between the bottom conditions in trenches within the same hadal province and their location along the bottom limb of the overturning circulation.





**Figure 9. Conditions in each trench at 7000 m depth of (a) In-situ Temperature (°C) (b) Conservative Temperature (°C) and (c) Absolute Salinity (g kg-1). JPT, Japan Trench, IOT, Izu-Ogasawara Trench, PHI, Philippine Trench, PAL, Palau Trench, YAP, Yap Trench, MAR, Mariana Trench, SCZ, Santa Cruz Trench, NHT, New Hebrides Trench, KRT, Kermadec Trench, TON, Tonga Trench, JAV, Java Trench, SAND, South Sandwich Trench.**



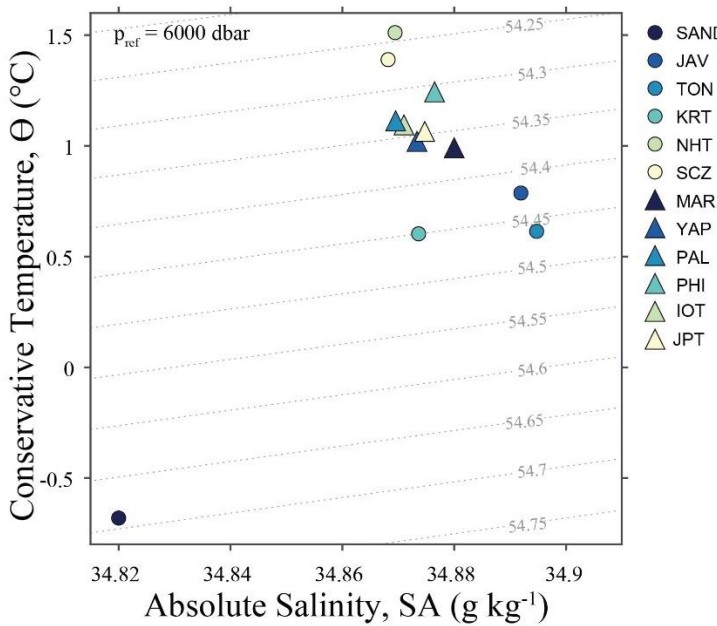

**Figure 10. Average bottom conditions in Θ-S$_A$ for each trench. Isopycnals (contour lines) show the potential density referenced to 6000 dbar. See Figure 9 for location abbreviations and Table 1 for bottom depth.**

### 3.3 Source water type analysis

Preliminary results indicated vertical variation in the water-mass fractions over the abyssal waters (4000 – 6000 m) and negligible vertical variation in the hadopelagic. Within abyssal waters, the KRT, the southernmost trench of the Pacific, exhibits the greatest proportion of AABW over depth (~0.49) (Figure 11a). The proportion is lower at 4000m for TON (~0.47) (Figure 11b); however, it has a comparable proportion of AABW to KRT throughout the hadopelagic (0.58) (Figure 12). MAR and YAP showed a similar proportion of AABW in abyssal waters (~0.19); however, MAR had a higher proportion of AABW in the hadopelagic (~0.22). The conditions in PAL provided a similar output of AABW proportions to IOT and JPT in both depth ranges. The proportion of AABW in the abyssal waters (~0.03) and the hadopelagic (0.05) were the lowest for PHI. There was a greater range uncertainty due to the source water types (Table 2) in the northern hemisphere than in the southern hemisphere. The range uncertainty is highest for PHI in the abyssal waters (Figure 11f) and the hadopelagic (Figure 12). The more considerable variation in LCDW source water types, compared to AABW (Table 2), had a greater impact on the range uncertainty overall in the northern hemisphere compared to the southern hemisphere. Generally, the proportion of AABW decreases as the distance from the source latitude (60ºS) increases, apart from PHI which exhibits conditions more similar to those of LCDW.





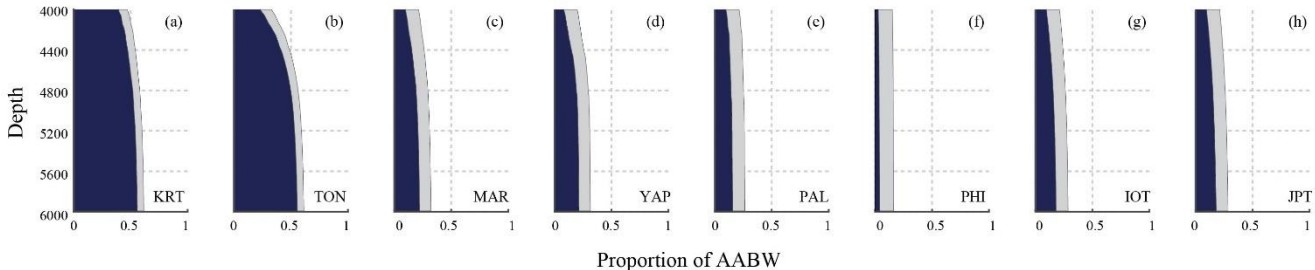

**Figure 11. Proportion of Antarctic Bottom Water (AABW) in the abyssal waters above (a) KRT, (b) TON, (c) MAR, (d) YAP, (e) PAL, (f) PHI, (g) IOT and (h) JPT in the blue shaded profile. The proportion of LCDW is 1 minus the proportion shown. The grey area represents the range uncertainty between possible source water type combinations of AABW and LCDW in the SWT analysis. The standard error of the mean was less than 10⁻⁴ in all instances.**

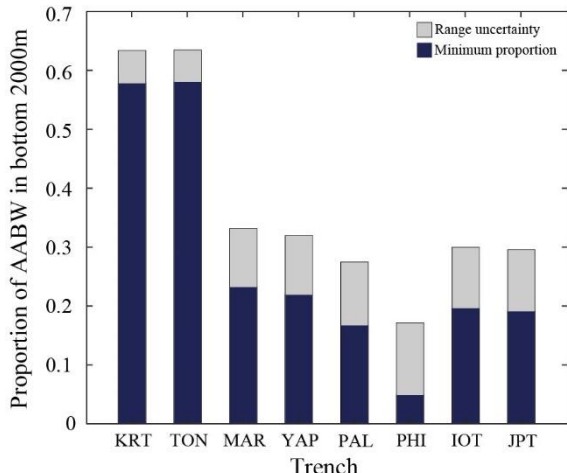

**Figure 12. Proportion of AABW in the bottom 2000 m of trenches in the western Pacific. The proportion of LCDW is the remainder of the proportion to equal 1. The range uncertainty is the mean range between possible source water type combinations of AABW and LCDW in the SWT analysis (Table 2). The standard error of the mean was less than 10⁻⁴ in all instances.**

## 3 Discussion

The variations in trench conditions were consistent within their hadal provinces and current knowledge surrounding bottom water circulation out of the Southern Ocean (Gordon et al., 2020; Naveira Garabato et al., 2002; Talley, 2013a), in the Indian Ocean (Wijffels et al., 2002; Wang et al., 2012; Mantyla and Reid, 1995) and throughout the western Pacific (Liu, 2022; Germineaud et al., 2021; Tian et al., 2021; Zhai and Gu, 2020).

At 7,000 m, the temperature is relatively consistent within the hadal provinces. The same patterns in differences are seen in T and $\Theta$ (Figure 9 a and b). These differences occur between either side of the South Fiji Basin, with TON and KRT being ~1°C cooler than NHT, SCB or SCZ at 6000 m. Surrounding the Philippine Sea, PHI on the eastern side is ~0.2°C warmer than MAR, YAP, and PAL to the southeast and IOT and JPT to the northeast. These two differences in the south and northeast Pacific are likely due to the moving of cold Lower Circumpolar Deep Water (LCDW) northward at hadal depths along the



Tonga-Kermadec Ridge and north-eastward to reach Mariana-Yap Junction (Kawabe and Fujio, 2010). The LCDW does not penetrate into the south Philippine Basin due to the Central Basin Fault, which may explain these differences and those seen in the SWT analysis (Tian et al., 2021). The same applies to NHT and SCZ, which are less ventilated by the LCDW

(Germineaud et al., 2021).

Our observations revealed fine-scale variations in the hydrographic profiles that require further exploration. The Java Trench, for which we believe is the first full-depth hydrographic profile, displays a $\Theta$-$S_A$ profile shape similar to Bayhaqi et al. (2018) and Atmadipoera et al. (2009). There is a split in the $\Theta$-$S_A$ shape at 0.8°C (Figure 3b), likely due to the opposing movement of bottom water on either side of the trench sills, with the northernmost profile displaying a slightly higher salinity (Zhai and Gu,

2020; Johnson, 1998). The northernmost profile displays a slightly higher salinity. A similar diversion appears in the Yap Trench (Figure 6b), with the deeper, saltier profiler nearer to the Yap-Mariana Junction, possibly due to local water isolation at hadal or bottom water depths, as suggested by Liu et al. (2022). Although the CTD profiles for IOT are more variable compared to neighbouring JPT profiles, they meet our quality control criteria. The IOT profiles display an increase in $S_A$ over a constant $\Theta$ of 1.06°C, like the profiles from section P09. The IOT's location within the primary flow path of AABW/LCDW

and UCDW may contribute to increased mixing (Zhou et al., 2022; Taira, 2006). Conversely, our observations at PHI show a discrepancy in $S_A$ to the WOCE deployments (Figure 8a), which could be due to temporal differences, conductivity measurement inconsistencies, or possible sensor drift (van Haren et al., 2021; van Haren, 2022; Gouretski et al., 2022). Further supporting this, moored profiles from (Wang et al., 2017), 4° south of our observation shows a lower average $S_A$ (Wang et al., 2017). Given these complexities, additional measurements within these trenches are necessary.


The warmer waters of PHI, and the resulting lowest fraction of AABW in the hadopelagic (0.05±0.01) compared to nearby trenches is consistent with the observed circulation within the Philippine Basin (Tian et al., 2021; Zhai and Gu, 2020; Wang et al., 2017). The bottom water enters through the Yap-Mariana junction, spilling into the Philippine Sea, flowing northward and westward, but blocked to the south by the Central Basin Fault (Tian et al., 2021). Aside from PHI, the proportion of AABW

within a trench decreases with increasing distance from the Southern Ocean (Figure 12). Overall, the proportion of AABW within trenches along the western Pacific is consistent with bottom water circulation in the Pacific (Kawabe et al., 2003; Kawabe, 1993). However, our water mass analysis results limited by the constraints placed on the water mass seawater properties used in the analysis. A more robust understanding of the filling and circulation deeper than 6000 m is rare and biased towards MAR (van Haren et al., 2017; Jiang et al., 2021), while circulation of abyssal waters above the trenches in the Pacific

are well described (Liu, 2022; Taft et al., 1991; Kawabe, 1993; Zilberman et al., 2020). Recent studies on PHI describe a cyclonic motion over the southern portion of the trench (Tian et al., 2021; Zhai and Gu, 2020; Wang et al., 2017), originally detailed by Johnson (1998) and present over other trenches (Mitsuzawa and Holloway, 1998; Warren and Brechner Owens, 1985; Huang et al., 2018; Nagano et al., 2013; Ma et al., 2021). Given the difficulties in deploying moored current profilers to hadal depths, where future deployments are limited to CTDs, deployments should be along transects in a meridional or zonal

direction to allow calculations of geostrophic velocities within and above the trench sills. Furthermore, deploying sensors with





higher frequency measurements than those included here is needed to uncover details on turbulent mixing processes and internal wave characteristics within weakly stratified trenches (van Haren, 2020a, 2023).

From around 9000 dbar, increases in $S_A$ of approximately 0.003 psu/1000dbar were observed in the KRT, TON, MAR, and PHI trenches. This was despite applying a linear pressure correction to the conductivity data during processing. Similar findings were noted by van Haren et al. (2021) and Taira (2005) in the Mariana Trench. Determining whether this increase was an instrument artifact, or a true salinity rise complicates water mass analyses. Following discussions with the CTD instrument manufacturer, SeaBird Electronics, we decided against implementing the corrections suggested by van Haren et al. (2021) for a comparable SBE 911 CTD. The decision was because corrected salinity data still exhibited an increase when the correction

was applied. Van Haren et al. (2021) proposed that this increase could signify dense modified AABW mixing with higher salinity LCDW in the trenches over long time scales. This observation is particularly relevant within KRT and TON, which serve as a conduit between LCDW and cold AABW from the south to the north of the Pacific (Reid, 1997; Zilberman et al., 2020). The varying rates of $S_A$ increase across the trenches suggest that the increase may not be solely due to instrumentation. Other potential causes include small-scale influences from diapycnal mixing and internal tides (van Haren, 2020a), or saline

pore water being expelled from deep-sea sediments due to intense pressures (Oguri et al., 2022; Glud et al., 2013; Turnewitsch et al., 2014), which could result in salinity spikes. Nevertheless, when examined from a broad-scale perspective, our SWT analysis indicates that this slight SA increase does not significantly affect the water mass proportion over hadal depths in these trenches (±4E10-5).

The observed latitude-dependent pattern is key to understanding climate change scenarios, especially alterations in AABW ranging from export rate to warming (Zhou et al., 2023; Jacobs et al., 2022; Bai et al., 2022; Boeira Dias et al., 2023). Warming has been predominant in the Indian Ocean, with contrasting freshening in the eastern region and increased salinity in the western region (Choi and Nam, 2022; Thomas et al., 2020). Similar trends of warming and freshening are well-documented in the Pacific (Purkey et al., 2019; Johnson et al., 2019; Lele et al., 2021) and the Atlantic (Campos et al., 2021; Liu and Tanhua,

2021). Compared to the 1990 WOCE S4, the South Sandwich trench profiles are fresher and warmer, a predictable discrepancy considering the time difference (Gordon et al., 2020; Zhang and Delworth, 2016). We intentionally did not address temporal changes in the WOCE sections, given their comprehensive coverage elsewhere (Hautala and Finucane, 2022; Wijffels et al., 2002; Naveira Garabato et al., 2014; Wijffels et al., 1998; Talley, 2013a). While single observations from some trenches provide a preliminary perspective, they should be interpreted with caution when drawing long-term warming or freshening

trends, even if it occurs in the overlying abyssal waters (Schmidt et al., 2023). Increased physical data from hadal depths, specifically regions other than the Mariana Trench, are required to understand long-term temporal and spatial differences thoroughly.

Bottom water masses, originating in the high latitudes, ventilate approximately 75% of the ocean below 1,500 m through
global overturning circulation (Robinson and Stommel, 1959; Stommel and Arons, 1959; Khatiwala et al., 2012). A trench's
conditions over hadal depths are significantly dictated by its geospatial position along the large deep circulation path,
introducing complexity and spatial heterogeneity (Kawabe and Fujio, 2010; Kawabe, 1993). Considering these factors, our
results are consistent with the current body of knowledge surrounding hadal trenches and add to our understanding.
Additionally, trench environments exhibit significant spatial heterogeneity of species, encompassing macrofauna such as
amphipods to microscopic life forms, including single-celled organisms and bacteria (Weston et al., 2022; Schauberger et al.,
2021; Jing et al., 2018). Intensifying our observational focus and understanding of the physical conditions in the hadopelagic
can offer deeper insights into ecological interconnections, diversity, and the driving forces behind such biodiversity. This
knowledge is particularly critical considering the vertical transport of organic material into these trenches' sills, a process
fundamental for life sustenance (Ichino et al., 2015; Glud et al., 2013). Thus, despite potential enhancements, the baseline
environmental conditions elucidated in the current study serve as an indispensable foundation for future research across
ecological, biological, or physical oceanographic understanding (Levin et al., 2019).

**Conclusion**

This study reinforces the importance of understanding hadal trench conditions and their heterogeneity due to their geospatial
positions along deep circulation paths. Distinct geographical trends were identified, with the South Sandwich Trench and Java
Trench exemplifying diverse profiles due to their respective locations. Vertical variations in water-mass fractions were
observed across trenches, with the southernmost Pacific trench, the Kermadec Trench, holding the most substantial Antarctic
Bottom Water proportion. Warmer waters in the Philippine Trench indicated reduced Antarctic Bottom Water presence,
consistent with its remote location on the western-most side of the Philippine Sea. A consistent salinity increase at depth across
several trenches was noted, possibly due to deep-sea sediment processes or water mass mixing. Latitude-dependent patterns
are crucial for understanding potential climate change scenarios and suggest consistent warming and freshening trends in the
Indian, Pacific, and Atlantic Oceans. The study provides valuable baseline conditions for future ecological, biological, and
physical oceanographic exploration and underscores the need for high-frequency sensor deployment and expanded trench
coverage.

**Code availability**

NA

**Data availability**

Data will be available through PANGEA



**Supplement link**

Link to supplementary pdf

**Author contribution**

JK completed the data processing, data analysis and prepared the manuscript with contributions from all co-authors. JK and JZ designed the data analysis. AJ collected the data. CP provided data presentation suggestions and edits.

**Competing interests**

The authors declare that they have no conflict of interest.

**Acknowledgements**

We thank the captain, crew, and company of the DSSV *Pressure Drop* during the 'Five Deeps' expedition and 'Ring of Fire' expeditions. We thank Victor Vescovo and Cassie Bongiovanni (Caladan Oceanic LLC, US), Patrick Lahey, Shane Eigler, Tim Macdonald (Triton Submarines LLC, US), Rob McCallum (EYOS Expeditions, UK). The authors would also like to thank Devin Harrison for the map in the Supplementary.

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
