# Peer review of "Water properties and bottom water patterns in hadal trench environments"

_EGUsphere, 2023_

## Author Response (AR1)

The authors would like to thank the two anonymous reviewers and the comments from the research community regarding this manuscript. The responses have been very inciteful and highlight the necessity for this research to be disseminated to the oceanographic community and the wider ocean science community.

Our responses to the comments below are in red.

**Reviewer #1 comments**

As I know, the CTD measurements have shown pressure dependency and hysteresis, with each probe exhibiting unique characteristics. While the authors have made efforts to correct this, such as the "linear pressure correction to the conductivity data" (L370) and offset correction for "the drift of conductivity measurement" (L177) using the WOCE/GO-SHIP dataset, there are lingering doubts about their sufficiency.

My concerns arise from the steep salinity increases close to the bottom in main figures and the substantial salinity deviations between corrected data and neighboring WOCE/GO-SHIP datasets seen in Figures S2 and S3. To address this, I guess it's crucial to compare vertical profiles among each CTD probe. Unfortunately, the main body of the manuscript lacks such profiles, with only limited data in Figures S2 and S3.

The phenomenon of increasing salinity in the hadal zone is fascinating. If confirmed, it could significantly contribute to oceanography and deep-sea biology. I remain hopeful that the data proves to be valid and reliable, considering its potential implications.

Thank you for your comments. Our caution stems from our lack of water samples over the profile, therefore no salinometer measurements, and the maximum GO-SHIP data depth of 6000 meters. We have added this detail specifically to the manuscript when discussing the increase (L100 and L401). We have also added further clarification on the correction methods (L185) and in the supplementary.

Some locations only had single measurements, and the T-S diagrams provided more insight, given the same data, we felt it unnecessary to put the profiles in the main body of the manuscript. The profiles are provided now in the supplementary results. Similarly, this increase in salinity can be seen in the TS diagrams and forms a part of the discussion, hence our omission of the same data in the main body of the manuscript.

Thank you for your excitement! We plan to obtain more hadal CTD measurements to prove or disprove this increase conclusively with repeat measurements, water samples, and other CTD manufacturers.

**Reviewer #2 comments**

This manuscript presents analyses of CTD data taken in and over deep ocean trenches by GO-SHIP repeat hydrographic cruises (incorrectly referred to as WOCE cruises in the manuscript) and by full-ocean-depth landers.  This is a revision of an earlier manuscript, and is much improved over that earlier version. It contains interesting new information, and should be suitable for publication following revision. Specific comments follow, indexed by line number, L, where applicable.

Thank you for the positive comment. Replies and revisions responding to each comment are in red.

1.Title.  Consider deleting "Examining baseline" in the title.

Thank you for the suggestion, this has been changed.

2.L15. Consider changing "Increases in salinity patterns" to "Salinity increases with increasing depth".

Changed to "Salinity increases with increasing depth for profiles over 10,000 dbar, with potential causes…"

3. L20-110 and discussion section.  The potential effects of geothermal heating on halal trench mixing (e.g., van Haren, 2023, Dynam. Atm.& Oceans) as well as T-S evolution (e.g., Joyce et al., 1986, Deep-Sea Res. A) and stability should be introduced here and then incorporated into the discussion.  Even the weak bottom heating in trenches will tend to cause convective turbulence in a (possibly quite thick) bottom mixed layer, working against establishment or maintenance of stabilizing deep salinity gradients potentially caused by other mechanisms.

Thank you for this comment. The third paragraph of the introduction has been changed to reflect these comments (L43) and it has been included throughout the discussion (L380).

"The extrapolation of hydrographic conditions to the broader physical oceanographic context of a trench system has been notably underrepresented in research. However, regional studies have provided insight into these depths but with some bias towards the *Challenger Deep* in the Mariana Trench (Greenaway et al., 2021; Mantyla and Reid, 1978; Taira et al., 2005; van Haren et al., 2017, 2021; Taira et al., 2004), and neighbouring

trenches (Kawagucci et al., 2018; Taira, 2006). Long-term temperature sensor deployments have shown the impact of internal tidal waves and turbulent spurs due to warm waters pushed from above the trench on the de-stagnation of the water below 6000m. These examples of turbulent mixing, excluding horizontal advection, may reduce the stratification on a similar order as geothermal heating (convective turbulence) (van Haren, 2023). Considering this, turbulence can be ten times higher in the upper hadopelagic (6,500 – 8,500 m) compared to the bottom of the hadopelagic (10,300 – 10,850 m) (Huang et al., 2018). Local cyclonic circulation over trenches has been identified over the Philippine Trench (Zhai and Gu, 2020; Tian et al., 2021) and the Mariana Trench (Huang et al., 2018), with both circulation patterns informed by bottom water circulation. "

4. L22. Consider changing "basins" to "a few deep basins".

Changed

5. L25. Consider changing "cool" to "cold", and is light penetration really "limited" at 6000 m? This reviewer would have thought it was effectively zero, although they are admittedly not an optical oceanographer.

L25. Changed to "cold temperatures and no light penetration..". Yes, there is no light therefore abyssal and hadal.

6. L28-29. This sentence is confusing and needs to be rewritten, perhaps split into two. "The 2-dimensional-area of the seafloor with depths greater than 1% (Harris et al., 2014)." is fine. However, what is meant by the second clause? Volume and depth are treated as somehow equivalent, which is confusing dimensionally and conceptually. At any rate, that second clause needs rethinking.

Changed to "Over 6,000 m, the volume is approximately 0.21% of the total ocean, however, it is 45% of the ocean's total depth range (Jamieson 2015)."

7. L50. Change "between" to "among".

Changed

8. L128. There is a grammatical error here that needs to be fixed so that readers can understand the meaning of this sentence.

Now L131. Changed to ""The Kermadec Trench connects at the southern end of the Tonga Trench within the same convergence system in the central South Pacific, separated only by the subducting Osborn Seamount (Jamieson et al., 2020). Deployments were made in

the Tonga Trench (~23˚S / 174˚W) to a maximum depth of 10,823 m, 9,986 m and in the Kermadec Trench (~32˚S / 177˚E)."

9. L137. Add a comma before the last and in the series.

Added

10. L177-181. The WOCE field program ceased circa 1998, 2000 at the latest. Repeat Hydrographic sections collected along historical WOCE lines within ±4 years of the lander expeditions would have been completed under the auspices of GO-SHIP (e.g., Sloyan et al., 2019, Frontiers in Marine Sci.). It would probably be useful to the reader to cite the years of the GO-SHIP sections used at each WOCE historical site. Also, how were the offsets determined? The optimal way would be to use conservative or potential temperature as the independent variable, and adjust the lander salinity to match the GO-SHIP CT-SA relation in a relatively stable portion of the water column (e.g., small lateral gradients and relatively slow circulation - likely the "oldest" deep waters rather than the more recently ventilated and presumably more variable bottom waters).

Yes, some have been incorrectly labelled as WOCE observations, while they are GO-SHIP observations. WOCE lines are included within the paper (see changes in Table 1 with distinctions). The distinction between the two has been made clearer including the appropriate referencing. The method you describe is how the offsets were calculated, this is detailed within the supplementary information (Supplementary 1). We have added a sentence in the methods for clarity as well (L184).

11. L203-15. OMP analysis typically takes advantage of a non-negativity constraint and requires the water mass fractions to add up to unity, both of which make the calculation better determined. It is not clear from this description that this was done. It probably should be, otherwise OMP should not be invoked. In Matlab the function lsqnonneg in the optimization toolbox would be useful for adding the non-negativity constraint. Also, was any weighting used, as customary in OMP? If not, please note that, and if so, please note what it was. If this is all too much, it would be fine to reframe the problem as simple end-member mixing in CT-SA space with two end-members, and not OMP at all, since it could be simplified to that if desired.

Thank you for this comment and suggestions. A non-negativity constraint was considered, however not applied since our results did not have any non-negative values. Weightings from 1-1, 5-1 and 10-1 were tested, typical of OMP analysis for CT-SA. However, the difference in the AABW fraction from using 1-1 to 10-1 was 0.0001, hence we omitted using weighting in our results. Additionally, the exclusion of a weighting tends

to the problem being a simple end-member mixing in CT-SA space as you have suggested. We have made this methodology and decision making clearer in the paper (L205).

12. L227 and following.  The discussion here mostly quotes gradients from 4000 dbar to the bottom of the profiles. This practice mixes the regions above and below the sill of the trench. Readers might be more interested in gradients from the sill depth (which should be estimated for each trench) and the bottom.  This would allow a focus on trench processes and dynamics, rather than mixing trench and deep water processes and dynamics.

Thank you for this comment. We have modified this throughout the results (Section 3.1) to emphasis the rate of change in temperature and salinity over 6000 dbar. In the Θ - $S_A$ figures we have included 4000 dbar and deeper due to the inclusion of GO-SHIP and WOCE data. Additional gradients are identified for the deeper trenches to show the rate of $S_A$ increase.

13. L239 (and elsewhere?).  Change "monotonously" to "monotonically". ;-)

Changed

14. Figures 2-8 and discussion. It is interesting that almost all the lander CTD profiles (with the exception of in the Japan Trench) exhibit increasing salinity with increasing pressure at high pressures (salty tails) with various amplitudes, whereas the GO-SHIP data "tails" are either absent or small (the P08 "fresh tails" are implausibly statically unstable, and are nearly within the ±0.002 PSS-78 instrumental uncertainty). All of the GO-SHIP cruise CTD data would be calibrated to bottle salinity data. In general that calibration would include a conductivity cell compressibility coefficient that was determined by least squares fitting along with other calibration coefficients for each co sensor used on that cruise (but maybe the P08 calibration didn't include that term?). So the correction would be specific to the cruise and the sensor.  It would often be different from the nominal correction (based on the compressibility of glass) that Seabird Scientific provides.  Certainly the CTDs used on Deep Argo floats have exhibited a noticeable artifact owing to this issue (Kobayashi et al., 2021, Prog. Oceanogr.)  In addition, it seems possible that under the truly extreme pressure experienced by the lander CTDs, some nonlinearity in the interaction between the glass co cells and their plastic protective jacket could come into play.  So without careful (e.g., done to GO-SHIP standards) bottle salinity analyses with multiple samples collected at a variety of pressures (from the trench sill to the bottom) this reviewer is quite skeptical regarding the salinity increases with increasing pressure reported by the lander CTDs. They could be real, but a more likely explanation is that they are an artifact owing to an incorrect coefficient, or even an inadequate model

(e.g., linear when it perhaps should be non-linear), used to correct for conductivity cell compressibility.  The discussion should probably reflect this perspective.

Thank you for this inciteful comment. Yes, we were also sceptical that this increase was real, however given the findings of van Haren et al. 2021 we thought to discuss the possibility that this was a true increase. Notwithstanding, we have added the details you have provided, particularly details from Kobyayshi et al. 2021 and reflected this information more clearly in the discussion (L385 – L404).

15. L339-340. There is a repeated phrase in here.   Please edit to remove the repetition.

Removed

16. P369 and elsewhere.  Practical Salinity is reported on the dimensionless Practical Salinity Scale of 1978 (PSS-78) and Absolute salinity has "units" of g/kg.  There is no such thing as "psu".  Please revise the manuscript throughout accordingly.

This is a typo. Throughout we are referring only to Absolute Salinity and not Practical Salinity. This has been removed and changed to g/kg. The differentiation is highlighted at L232.

17. L378. The varying rates of salinity increase with increasing pressure could easily be solely due to instrumentation.  The same co sensor used on different cruises can require different compressibility correction coefficients as it ages.

This has been reflected throughout this paragraph, also encompassing the points you have made above in no 14.

---

## Author Response (AR2)

Response to reviewer in red
This manuscript presents analyses of CTD data taken in and over deep ocean trenches by WOCE and GO-SHIP hydrographic section and by full-ocean-depth landers. This revised manuscript adequately addresses the comments raised in the first review. A few, mostly minor suggestions follow, indexed by line number, L, where applicable.

1. L22. Consider changing to "There remote regions are arguably..."
Changed

2. L28. Consider changing to "The volume of water with z > 6000 dbar is approximately 0.21% of the..."
This has been changed to:
"The volume of water with depth greater that 6,000 m, is approximately 0.21% of the total ocean, however, it is 45% of the ocean's total depth range (Jamieson, 2015)."

3. L98. Change "and early 2000s" to "and 1990s". The WOCE observation period officially stopped in 1998. Sections occupied after WOCE and before GO-SHIP were under the ageis of CLIVAR/CO2.
Changed

4. L202-212. The use of a non-negativity constraint does theoretically improve the results for OMP even if the results are positive without it. However, with just T-S variations, 1-1 weighing, and 2 end-members, this analysis isn't really OMP, it's just straight-up end-member mixing, right? Please consider revising the text to reflect that, if you agree.
Yes, we agree, we have modified the starting sentence of the paragraph to make this clear.
"We apply an end-member mixing analysis to the profiles along the western Pacific"

5. L290. Change "statistically" to "statically".
Thank you for picking this up. This has been changed also in the caption for Figure 8.

6. L362. It seems highly unlikely that temporal variations play a substantial role here in the T-S differences observed here (and elsewhere in the report). The deep, abyssal, and hadal waters are quite spatially homogenous, and quite old, in the region. A big offset in salinity here is almost certainly owing to a calibration issue, and not temporal variability.
You are correct. Previously this was referring to the full water column, hence 'temporal changes' were appropriate for the upper waters. "Temporal differences" has been removed.

7. L390. Calibrations for WOCE and GO-SHIP cruises certainly often require departures from the "nominal" co cell compressibility coefficient to match CTD and bottle salinity measurements.
Yes correct. We have modified this sentence to reflect the nuance of CTD corrections specific to the instrument.
"The decision was because corrected salinity data still exhibited an increase when the correction was applied, and the conductivity correction is specific to the instrument."

8. L401. Change "data is" to "data are".
Changed